# Self-heating hotspots in superconducting nanowires cooled by phonon black-body radiation

Andrew Dane [1] ✉, Jason Allmaras[2,3], Di Zhu [1], Murat Onen[1], Marco Colangelo [1], Reza Baghdadi[1], Jean-Luc Tambasco [1], Yukimi Morimoto [1], Ignacio Estay Forno[1], Ilya Charaev [1], Qingyuan Zhao[1], Mikhail Skvortsov [4], Alexander Kozorezov[5] & Karl K. Berggren [1] ✉

Controlling thermal transport is important for a range of devices and technologies, from phase change memories to next-generation electronics. This is especially true in nano-scale devices where thermal transport is altered by the influence of surfaces and changes in dimensionality. In superconducting nanowire single-photon detectors, the thermal boundary conductance between the nanowire and the substrate it is fabricated on influences all of the performance metrics that make these detectors attractive for applications. This includes the maximum count rate, latency, jitter, and quantum efficiency. Despite its importance, the study of thermal boundary conductance in superconducting nanowire devices has not been done systematically, primarily due to the lack of a straightforward characterization method. Here, we show that simple electrical measurements can be used to estimate the thermal boundary conductance between nanowires and substrates and that these measurements agree with acoustic mismatch theory across a variety of substrates. Numerical simulations allow us to refine our understanding, however, open questions remain. This work should enable thermal engineering in superconducting nanowire electronics and cryogenic detectors for improved device performance.

Superconducting nanowires are the basis for a number of quantum technologies including single-photon detectors[1] and associated readout devices[2], as well as quantum phase-slip junctions[3]. However, the study of the thermal properties of superconducting nanowires is often incidental, despite the potential for radically altered heat transfer in nanoscale systems[4,5]. In phase-slip junctions, heating is detrimental to coherent tunneling of phase slips, though thermal runaway may enable the observation of single tunneling events[6]. Superconducting nanowire single photon detectors (SNSPDs)[1] on the other hand, rely on the creation of localized, photo-induced normal regions, or hotspots, in order to detect infrared photons[7–9]. Energy deposited into the SNSPD is eventually released into the substrate in the form of phonons. Macroscopically, the net rate of phonon emission is quantified by the thermal boundary conductance (TBC)[10–12] between the wire and substrate.

The TBC between SNSPDs and dielectric substrates is one of the main determinants of the maximum (non-latching) device speed[8], affects quantum efficiency[13], jitter and latency[14], and it may affect

[1]Department of Electrical Engineering and Computer Science, Massachusetts Institute of Technology, Cambridge, MA 02139, USA. [2]Department of Applied Physics and Materials Science, California Institute of Technology, Pasadena, CA 91125, USA. [3]Jet Propulsion Laboratory, California Institute of Technology, Pasadena 91109 CA, USA. [4]L. D. Landau Institute for Theoretical Physics, Chernogolovka 142432, Russia. [5]Department of Physics, Lancaster University, Lancaster, UK. ✉e-mail: aedane@mit.edu; berggren@mit.edu

observed switching currents, and dark count rates. At the early stages of photo-detection, the energy from a photon absorbed in a SNSPD is divided among a small number of quasiparticle and phonon excitations[15]. Pair-breaking phonons that escape into the substrate reduce the energy available to disrupt the superconducting state. Thus, a lower TBC may increase device detection efficiency[13]. On the other hand, pair-breaking phonons reflected at the substrate interface could lead to switching of the device at a later time, increasing the latency and jitter of the device[14]. After hotspot formation, electro-thermal feedback on its growth determines the conditions under which a device can operate in a free running mode, or whether it will latch into a resistive state; faster detectors require increased TBC[8]. Finally, as device DC bias currents are ramped up to increase detection efficiency and lower jitter, eventually vortices are drawn across the wire[16]. Vortex crossing releases energy which can lead to localized heating and increased vortex flow, and at sufficiently high currents, thermal runaway and hotspot formation[6]. Increased cooling could help reduce dark counts at a given DC bias. At sufficiently high dark count rates, the wire switches to a stable resistive state, well before the theoretical depairing current[17]. Seemingly all SNSPD performance metrics relate to how quickly heat is removed from the nanowire. Despite this, the TBC between SNSPDs and substrates has only been studied in a handful of cases[18,19].

Here, we attempt to quantify the TBC between superconducting nanowires and substrates using measurements of the current needed to sustain a hotspot inside the nanowire, known as the self-heating hotspot current ($I_{hs}$)[20]. Measurements of this type have been attempted previously[7], though usually with micrometer scale devices, one substrate type, and without clear comparisons to theoretical expectations[21,22]. In fact, based on our results, some of the TBC values reported in the literature appear to be larger than can be explained by theory[21,23], while other results can be re-interpreted within our scheme to good agreement[22]. In order to measure the TBC between nanowires and substrates, and clarify the description of self-heating hotspots in nanowires, we compare measurements of $I_{hs}(T_b)$ for 17 NbN nanowires across six different substrate materials, with analytical calculations and finite element electro-thermal simulations. Importantly, our method is applied to nanowire devices made from the same materials and using the same designs and processes as a number of state of the art superconducting nanowire single photon detectors. Our method requires no specific device designs or experimental setup beyond what is typically used in SNSPD measurements.

## Results and discussion

In order to make sense of our experimental results, we first review a theoretical model used extensively to interpret similar measurements. The Skocpol Beasley Tinkham (SBT) model[24], including later generalizations[25–27], forms the starting point for understanding hotspots in superconducting nanowires. The model considers a one-dimensional normal domain inside of an otherwise superconducting wire, centered at $x = 0$ and extending to $\pm x_N$. This one-dimensional model is appropriate when the width and thickness of the wire are narrower than the thermal healing length ($\eta$)[24], a parameter which governs how quickly the temperature profile can change along the wire. Although the ~5 nm thickness of our samples is always much less than the expected $\eta$, for some of our measurements the width was not strictly less than $\eta$ (see Supplementary Table 6). Edge effects which weaken superconductivity[28] are not thought to affect the following analysis, unless they produce an insulating state, in which case they may effectively narrow the wire. In addition, it should be noted that the width and thicknesses of the nanowires in this study are much less than the penetration depth[29], and as a result the current density is uniform across the cross

section of the wire. The heat balance equations in the normal and superconducting regions are given by:

$$-(K_N \cdot w^2 d)\frac{\partial^2 T}{\partial x^2} + \beta w^2 (T^n - T_b^n) = I^2 R_n \quad (|x| < x_N) \quad (1)$$

$$-(K_S \cdot w^2 d)\frac{\partial^2 T}{\partial x^2} + \beta w^2 (T^n - T_b^n) = 0 \quad (|x| > x_N) \quad (2)$$

where $K_N$ and $K_S$ are the thermal conductivities of the wire in the normal and superconducting states, respectively. $w$ and $d$ are the width and thickness of the wire, $I$ is the current, $R_n$ is the sheet resistance in the normal state, $\beta$ is a generic thermal boundary conductance with units of W/m²K$^n$, and $T_b$ is the bath/substrate temperature. The temperature at $\pm x_N$ is assumed to be $T_c$. The temperature profile along the wire, $T(x)$, is solved for assuming that the heat flow at $\pm x_N$ is continuous, and $T \to T_b$ as $x \to \pm \infty$. For a given $x_N$, the solution to Eq.s (1) and (2) determines a current-voltage pair. By solving for a range of $x_N$, we can trace out a current-voltage (IV) curve that contains a distinct region of near-constant current for a range of voltages. This current is the hotspot current, $I_{hs}$.

When the hotspot is sufficiently long, we expect that the temperature around $x = 0$ will be nearly constant, such that $\frac{\partial^2 T}{\partial x^2} \approx \frac{\partial T}{\partial x} \approx 0$[26], allowing us to drop the first term on the left hand side of (1):

$$\beta w^2 (T_{hs}^n - T_b^n) = I_{hs}^2 R_n \quad (3)$$

Here, $T_{hs}$ is the hotspot temperature, the nearly constant temperature in the normal domain near $x = 0$. $I_{hs}$ is the current required to maintain the hotspot via Joule heating. This relation was found by Dharmadurai and Satya Murthy (DSM)[26] to outperform their own more sophisticated attempts to model $I_{hs}$ data for long superconducting wires with ~mm widths and 10–20 nm thicknesses, originally measured in ref. 25. Interestingly, these were Al samples evaporated in an oxygen atmosphere, with resistivities in the range of recently reported granular aluminum[30]. DSM assumed that $T_{hs} = T_c$ for all $T_b$. This assumption is not compatible with (1), which predicts that $T_{hs}$ is a function of $T_b$, with $T_{hs}(T_b = 0) = T_c(1 + 1/n)$[24–26].

The value of the exponent $n$ in the second term on the left-hand side of (1) is an important parameter that captures the relevant physics and dimensionality of the excitations which cool the hotspot. For instance, for clean bulk metals, $n = 5$ would be appropriate if the bottleneck to heat flow was between electrons and phonons within the wire[31,32] while $n = 4$ could describe the same for a clean metal membrane[33]. For highly disordered bulk metals, $n = 6$ for electron-phonon coupling has been reported, consistent with theoretical predictions[34–36]. Non-integer values of $n$ may occur due to the roughness or structure of the interface that makes transmission wavelength-dependent[31,37]. Importantly, when the difference between the hotspot temperature and the bath is small, this term can be linearized to a term of the form $n\beta T_b^{n-1} w^2 (T - T_b)$, as is often done[10]. The additional terms in the prefactor are combined into a linear heat transfer coefficient, $\alpha = n\beta T_b^{n-1}$.

The original SBT paper[24] used a linearized TBC which was quickly shown to be invalid at low $T_b$[25]. The SBT model with linear TBC predicts that $I_{hs}(T_b) \propto \sqrt{T_c - T_b}$. This prediction has been used in the past to fit $I_{hs}(T_b)$ data to estimate $\alpha$[22]. When we square the hotspot current predicted by the original SBT, we find it is linearly related to the bath temperature, $I_{hs}^2(T_b) \propto (T_c - T_b)$. Written in this way, the deficiency in the original SBT model is made clear, when compared with experimental results[25,26]. However, this linearization was incorporated into initial descriptions of SNSPDs[1], without being addressed as such.

Figure 1 shows an example of our DC electrical measurements of $I_{hs}$ at various $T_b$, which we use to illustrate that the linearized SBT

model is not consistent with experiment, and points out open questions related to properly explaining $I_{hs}$. IV curves like the one shown in Fig. 1b were measured vs bath temperature ($T_b$) for 17 NbN nanowires across six different substrate materials. $I_{hs}$ is identified as the constant current 'plateau'[1] at non-zero voltages, observed when ramping the bias down after switching. Unlike the switching current ($I_{sw}$), $I_{hs}$ is deterministic and immune to noise in this configuration[38]. The measurement circuit is shown inset. IV curves were measured at 0.2 K increments of $T_b$, up to 12 K, and the resulting $I_{hs}(T_b)$ and $I_{sw}(T_b)$ for one device on Si is plotted in Fig. 1c. When $I_{sw}(T_b) > I_{hs}(T_b)$, the IV curve is hysteretic[39], and while non-hysteretic IV curves may contain evidence of hotspot formation[40], we limit our identification of $I_{hs}$ to temperatures where the IV curve is hysteretic and the presence of the hotspot unambiguous. If we square the $I_{hs}(T_b)$ data from Fig. 1c, we can immediately see that it is inconsistent with the predictions of the linearized SBT model, as shown in Fig. 1d. In Fig. 1e we plot $T_{hs}(T_b)$ at the center of a long nanowire numerically calculated using Eq. (1) with $K_N = K_S$ for $n = 1 - 5$, to illustrate a potential issue with assuming that $T_{hs} = T_c$ for all $T_b$, as done by DSM. Lastly, in Fig. 1f we plot the expected energy content for phonon modes in NbN at 10 K vs the wavelength of that mode, assuming that the NbN is three-dimensional. The vast majority of the energy in such a system is carried by phonons with wavelengths exceeding the 5 nm film thickness typical for SNSPDs, implying that the correct description of the phonon system is not the simple three-dimensional picture. Thus, we might expect to need to modify both the description of the electron-phonon coupling inside of the wire[41], as well as the phonon-phonon coupling across the material boundaries[42].

All of our hotspot current data can be explained by a thermal boundary conductance that is determined by phonon black body radiation into the substrate, and as a result, the phonon-emissivity between the nanowire and substrate is the primary factor in determining $I_{hs}(T_b)$. Equation (3) with $n = 4$, $I_{hs}^2 R_n = \beta w^2 \left( T_{hs}^4 - T_b^4 \right)$, equates the Joule heating in the hotspot with the net heat flow out of the wire due to a detailed balance of phonon flux at the interface. This value of $n$ is appropriate for three-dimensional phonons, and despite our suggestion that the thickness of the wire may alter the dimensionality, we find that we can fit all of our data using Eq. (3) with $n = 4$, and that the fitting parameters match calculated values. The right-hand side has the same form as the net radiative heat transfer from a black body at temperature $T_1$ to another at $T_2$ due to the Stephan-Boltzman law for blackbody radiation of photons, $P_{1 \rightarrow 2} = \sigma \epsilon A \left( T_1^4 - T_2^4 \right)$, where $A$ is an effective surface area, $\epsilon \leq 1$ is an effective emissivity, and $\sigma$ is the usual Stephan-Boltzmann constant. In the phonon case, $\sigma$ becomes $\sigma_{ph} = \frac{\pi^2 k_B^4}{120 \hbar^3} \left( \frac{1}{c_{L1}^2} + \frac{2}{c_{T1}^2} \right)$[10], where $c_{L1}$ and $c_{T1}$ are the longitudinal and transverse phonon velocities in one of the materials. For an average velocity, such that $\left( \frac{1}{c_{L1}^2} + \frac{2}{c_{T1}^2} \right) = \frac{3}{c_{avg}^2}$, we can write $\sigma_{ph} = \frac{\pi^2 k_B^4}{40 \hbar^3 c_{avg}^2}$ as done in ref. 43. Note that the present description is equivalent to the well-known Kapitza resistance at solid-solid interfaces. However, 'Kapitza resistance' is often used ambiguously, or in situations where small temperature differences invite linearization. Here, linearization does not work, as shown in Fig. 1d. We therefore avoid 'Kapitza resistance' in favor of 'phonon black body radiation' similar to ref. 43.

In order to compare experimental data with theoretical expectations, we calculated the phonon-emissivity between NbN and various substrate materials, using the acoustic and diffuse mismatch models. In the acoustic mismatch model (AMM) the interface between materials is assumed to be ideal, and incident phonons are transmitted and reflected specularly, in proportions that satisfy continuum mechanics

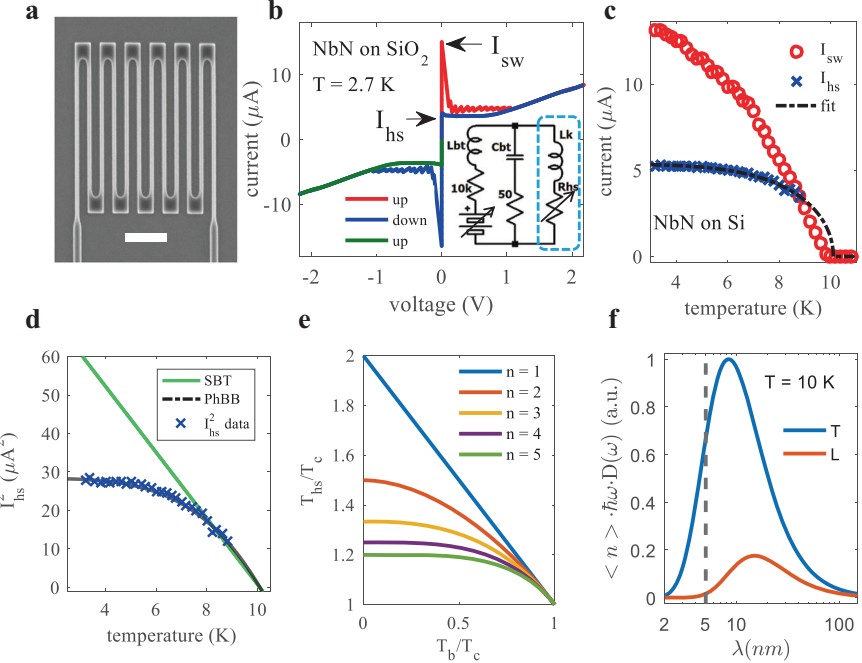

**Fig. 1 | Measurements and modeling of $I_{hs}(T_b)$. a** Scanning electron micrograph of a device measured for this work. Scale bar is 1 μm. **b** Typical hysteretic IV curve of a superconducting NbN nanowire with the switching ($I_{sw}$) and hotspot ($I_{hs}$) currents labeled. The measurement circuit is inset. Only the nanowire was cooled (blue dashed box). **c** $I_{sw}(T_b)$ and $I_{hs}(T_b)$ for NbN on Si device. Fitting of $I_{hs}$ is shown and discussed below. **d** The $I_{hs}(T_b)$ data and fit from **c** was squared and re-plotted. Displayed this way, it is easy to see the failure of the linearized SBT model to capture the shape of the data. **e** Calculated hotspot temperature ($T_{hs}$) as a function of $T_b$ and the exponent $n$ which describes the power law cooling to the substrate. **f** Normalized energy density carried by three-dimensional longitudinal (orange) and transverse (blue) phonons in NbN as a function of wavelength at 10 K, using the Debye model for the phonon density of states. Both curves have been normalized by the maximum of the transverse curve. Typical film thickness (5 nm) indicated by dashed line.

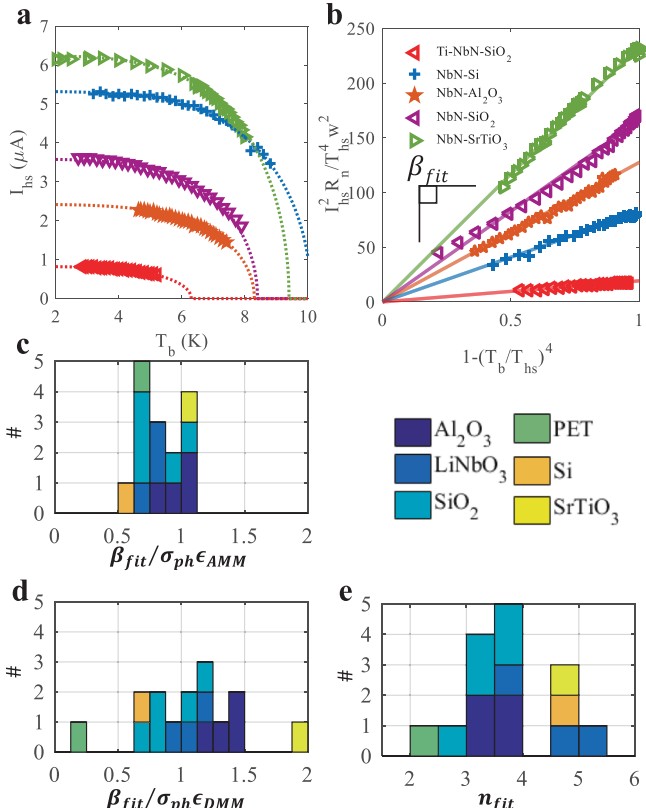

**Fig. 2 | Comparing the extracted phonon emissivity to AMM and DMM. a** $I_{hs}(T_b)$ for representative nanowires. Equation (3) is fit to the data using $\beta$ and $T_{hs}$ as fitting parameters with $n = 4$. Legend is shared with **b**. **b** Transformation of data in **a** allowing us to compare wires with different widths and fitted $T_{hs}$. Plotted this way, the fit curves become lines with slope $\beta_{fit}$. **c**, **d** each histogram entry is colored to indicate substrate. (c) Histogram of $\beta_{fit}/(\sigma_{ph}\epsilon_{AMM})$. **d** Histogram of $\beta_{fit}/(\sigma_{ph}\epsilon_{DMM})$. **e** Histogram of $n_{fit}$ that results from refitting all $I_{hs}(T_b)$ while allowing $\beta$, $T_{hs}$, and $n$ to vary.

at the interface[44]. In the opposite limit, the diffuse mismatch model (DMM) assumes that the interface is completely diffusive; incident phonons lose all memory of the direction of propagation before impinging on the interface. Phonons that arrive at the interface are scattered to one or the other side in proportion to the phonon density of states on either side of the interface. Mathematically, the emissivity values calculated using these two models are:

$$\epsilon_{AMM} = \left(\frac{\eta_L}{c_{L1}^2} + \frac{2\eta_T}{c_{T1}^2}\right)\left(\frac{1}{c_{L1}^2} + \frac{2}{c_{T1}^2}\right)^{-1} \tag{4}$$

$$\epsilon_{DMM} = \left(\frac{1}{c_{L2}^2} + \frac{2}{c_{T2}^2}\right)\left(\frac{1}{c_{L1}^2} + \frac{2}{c_{T1}^2} + \frac{1}{c_{L2}^2} + \frac{2}{c_{T2}^2}\right)^{-1} \tag{5}$$

$0 \le \eta_L \le 1$ and $0 \le \eta_T \le 1$ are the angle-averaged transmission factors for phonons originating in material 1, incident on the interface with material 2, for longitudinal and transverse phonons. We calculated $\eta_i$ using literature values for material properties (see Supplementary Table 4) while accounting for possible total internal reflection, as described by Kaplan[45]. These calculations are discussed in detail in appendix A of ref. [46]. We verified our AMM and DMM calculations by recreating table II from ref. [10]. In addition, we tabulated values of $\sigma_{ph}\epsilon_{AMM}$ for a variety of superconductor-substrate pairs that

may be useful for SNSPD applications. The results are given in Supplementary Tables 2 and 5, respectively.

In Fig. 2, we summarize our fitting of $I_{hs}(T_b)$ data using Eq. (3), and compare the extracted phonon emissivity to the predictions of AMM and DMM. We focus primarily on the case of $n = 4$, using $\beta$ and $T_{hs}$ as fitting parameters. In Fig. 2a, a representative subset of the $I_{hs}(T_b)$ data and fit lines are shown. Measured values of $R_n$ at 12 K, and measured or design values of $w$ were input into the fitting expression. All but one fit had an adjusted $R^2 > 98\%$. The adjusted $R^2$ for NbN on polyethylene terephthalate (PET) was 94%. Data and fit lines from Fig. 2a are replotted in Fig. 2b in order to more easily compare wires of different width and $T_c$ value and more clearly illustrate the observed range of $\beta_{fit}$ values. Figure 2c, d are histograms of the fitting parameter $\beta_{fit}$ divided by calculated values of $\sigma_{ph}\epsilon_{XMM}$. Entries into the histogram are colored according to the substrate material. In these histograms, we have excluded the two NbN-Ti bilayer on $SiO_2$ devices, which appear to have partially alloyed during the fabrication process, making it difficult to describe them acoustically (see Supplementary Fig. 3). While the values of $\beta_{fit}$ appear systematically lower than $\sigma_{ph}\epsilon_{AMM}$, our data suggests that AMM captures the essential physics better than DMM. This is consistent with recent work where the value of the thermal boundary conductance consistent with measurements was about 20% of the value calculated using DMM[47]. This analysis also worked for data extracted from ref. [22], which erroneously followed the linearized SBT model, yielding $\beta_{fit}/\sigma_{ph}\epsilon_{AMM} = 0.72$ (see Supplementary Fig. 2).

Deviations from the fit shape were noticeable in some data sets. In Fig. 2a, b, for instance, in our NbN on $SiO_2$ data (purple triangles) and in our NbN on polyethylene terephthalate (not shown; to be discussed at length elsewhere). The substrate we call $SiO_2$ was a 300 nm thick silicon thermal oxide layer grown on top of Si. The additional phonon scattering from the $SiO_2$-Si boundary, as well as high intrinsic scattering in the amorphous $SiO_2$[19,48], may push these samples away from the phonon radiation regime and cause the observed shape deviation. This may also explain why we predict (measure) a higher $\sigma_{ph}\epsilon_{AMM}(\beta)$ for NbN on $SiO_2$ than $Al_2O_3$, and yet NbN SNSPD count rate measurements suggest higher TBC on $Al_2O_3$[49]. Prompted by these deviations in shape, we repeated all fits, allowing $n$ to vary in addition to $\beta$ and $T_{hs}$. The resulting values of $n$ are plotted as a histogram in Fig. 2e. While this histogram gives the impression that a number of our samples may be described by a cooling mechanism with $n = 5$, we find from our simulation work that the process of extracting $n$ from $I_{hs}(T_b)$ data is likely unreliable.

To support and extend our analysis, we used a one-dimensional finite element simulation to repeat the present experiments in-silico. This simulation primarily follows ref. [9] and consists of a set of coupled heat balance equations that determine the electron ($T_e(x)$) and phonon ($T_{ph}(x)$) temperatures inside the hotspot, incorporating effects of the local superconducting gap $\Delta(T_e(x))$. Allowing for two temperatures inside of the hotspot is in contrast to the analytical models mentioned so far and our foregoing discussion, which tacitly assumed $T_e = T_{ph} = T_{hs}$. The temperature of the electron subsystem at every point in the simulation is governed by heat diffusion among electrons, Joule heating due to the hotspot current, and coupling between electrons and phonons proportional to $T_e^5 - T_{ph}^5$. Similarly, the local temperature of the phonon system is a balance between phononic heat conduction along the wire, heat received from the electron subsystem, and heat that escapes to the substrate described by a term proportional to $T_{ph}^4 - T_b^4$. As our model was originally setup for time-dependent simulations of photo-detection, the natural way for us to describe the coupling strengths was as timescales. The electron–phonon coupling strength is quantified in our simulation by a characteristic timescale for energy exchange, $\tau_0$[50], with estimates ranging between 200 and 1000 ps for NbN[9]. Similarly, the phonon escape time, which is another way of describing the TBC, we estimate

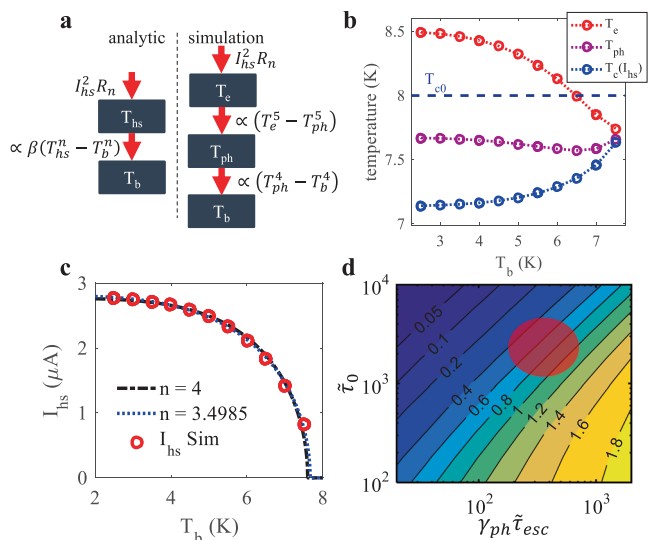

**Fig. 3 | Electro-thermal simulations of hotspots in superconducting nanowires.**
**a** Illustration of the conceptual difference between the analytic model, and numerical simulation. **b** Simulated electron ($T_e$) and phonon ($T_{ph}$) temperatures in the center of a hotspot, as a function of $T_b$, along with $T_c(I_{hs})$. **c** Simulated $I_{hs}(T_b)$ (red circles), along with fits using Eq. (3) with $n = 4$ (black), and $n$ as free parameter (blue). **d** Heat map of $\beta_{fit}/\beta_{sim}$ as a function of a characteristic electron-phonon coupling time ($\tilde{\tau}_0$), and a scaled phonon escape time ($\gamma_{ph}\tilde{\tau}_{esc}$). The red ellipse indicates where we expect the values for NbN to fall based on literature and our measurements.

to be between 50 and 250 ps. Lastly, the boundaries of the hotspot separate the superconducting and normal phases, and this boundary is dependent on both temperature and current[51]. To account for this in our model, the local critical temperature is reduced by the presence of current; we invert the Bardeen equation for the depairing current as a function of temperature, to give us an equation for the $T_c$ as a function of current. Further details of the model are given in the Supplementary Methods.

Figure 3 summarizes the results of our simulation efforts which included: (1) calculating the temperatures of the electron ($T_e$) and phonon ($T_{ph}$) sub-systems in the center of a stable hotspot, (2) recreating the measurement and fitting procedure with simulated data. Figure 3a illustrates the conceptual difference between how our fitting equation treated the hotspot (left), and how the simulation did (right), with red arrows indicating the net power flow. In Fig. 3b we report the simulated $T_e$, $T_{ph}$, and the effective $T_c$ in the center of a simulated hotspot, as $T_b$ is varied. As $T_b$ is reduced, $T_e$ increases in a manner qualitatively similar to what SBT predicts for $T_{hs}$ (see Fig. 1e), with a reduced magnitude. $T_{ph}$ is bounded above by $T_e$ and below by $T_b$, but it's exact value depends on the relative strengths of the electron-phonon coupling in NbN, and the phonon-phonon coupling across the wire-substrate boundary. For NbN, we find that $T_{ph}$ is nearly constant across $T_b$. This helps explain why we can use one temperature to fit our $I_{hs}$ data. In Fig. 3c we fit the simulated $I_{hs}(T_b)$ data with the phonon blackbody radiation expression. When we set $n = 4$, and allow $\beta$ and $T_{hs}$ to vary, the best fit value of $\beta$, $\beta_{fit} = 528$ W/m²K⁴, is within 2% of the value entered into the simulation beforehand, $\beta_{sim} = 518$ W/m²K⁴. When we allow $\beta$, $n$ and $T_{hs}$ to vary, the best fit $n$ is close to 3.5. Based on these simulation results, our fitting procedure seems unreliable in determining $n$.

In order to better understand how much the material properties of the film and substrate might influence the accuracy of our method, we repeated our simulation and fitting procedure while varying the strength of the electron-phonon coupling in the wire and the thermal boundary conductance to the substrate. The results are reported in

Fig. 3d. By varying the characteristic time, $\tilde{\tau}_0 = \frac{\tau_0}{\hbar/k_B T_c}$[50], as well as a scaled phonon escape time, $\gamma_{ph}\tilde{\tau}_{esc} = \frac{\gamma_{ph}\tau_{esc}}{\hbar/k_B T_c}$, we could probe different electron-phonon coupling strengths and thermal boundary conductances, within our established model[14]. This method follows ref. 9, where $\gamma_{ph}$ is the ratio between the electron and phonon specific heat at $T_c$, with further details given in the Supplementary Methods. In Fig. 3d we plot a heat map of $\beta_{fit}/\beta_{sim}$ vs $\tilde{\tau}_0$ and $\gamma_{ph}\tilde{\tau}_{esc}$. The red ellipse approximates the region where we would expect to find our NbN, based on published and measured data. In this region $\beta_{fit}/\beta_{sim}$ takes on values that are similar to what we found experimentally (see Fig. 2c), roughly from 0.4 to 1.2. We take this as an indication that our simple measurement and fitting procedure does as well as theoretically possible, and that when combined with knowledge of the electron-phonon coupling strength in a given superconducting material, could be used to improve the accuracy of this method.

Despite decades of progress, the acoustic and diffuse mismatch models often fail to accurately predict the observed thermal boundary conductance, which is why we consider our results to be highly encouraging. An agreement of ~20% between measurement and calculation is looked upon favorably[39]. In comparison, the values we extract experimentally are significantly closer to the values expected from AMM, despite many simplifying assumptions. The reason for this degree of agreement is still unclear. It may be that the stringent requirements put on the material in order to form state of the art SNSPDs also improves their performance in this application. High performance SNSPDs are fabricated from material that is flat, smooth, and without significant variation in the material properties along the wire. Sputtering ensures that the material is well bonded to the substrate[44]. The result may be a device that is ideally suited for nanoscale thermal transport measurements of this kind.

Of course, there are still open questions about the degree of phonon localization in the wire, and about the description of electron–phonon coupling in nanoscale dirty metals[45]. Others have reported a discrepancy between the expected phonon dimensionality, and that which best described heat transfer in short and thin superconducting nanowires[51]. One possible solution, based on a simple model of phonon quantization in thin films[42], is that an altered phonon density of states leads the phonon subsystem to slightly overheat for a given energy flow vs what we expect. This overheating would result in shorter dominant phonon wavelengths, which would then make the 3D picture we have used, more plausible, without invalidating our analysis. Likely, the proper solution to this problem requires detailed modeling of individual wires as partially clamped plates which can host a variety of surface and guided phonon modes[52], while also self-consistently accounting for variations in electron–phonon coupling due to reduced phonon dimensionality[53–55]. Such variations result in changes to the exponent which describes the power law coupling between the electron and phonon temperatures. As we showed in our simulation work, even when we prescribed the exponent values beforehand, our method did not reliably extract them afterward, which is why we do not take the data in Fig. 2e as evidence of either reduced phonon dimensionality or altered electron-phonon coupling. However, the good match between our experimental results, analytic calculations and simulated results suggest that overall this model captures the essential physics, and that additional effects should be refinements of the current model.

Our work can provide guidance for thermal engineering of next-generation SNSPDs. For NbN SNSPDs, reducing β by choice of substrate should lead to improved detection efficiency at longer photon wavelengths[7]. Increasing β has already been shown as a promising route for improving the bandwidth of NbN hot electron bolometers[46], and phonon trapping has been shown to improve

single-photon energy resolution in kinetic inductance detectors[47]. For most of the device performance parameters, the degree of impact is still not well understood, and sophisticated modeling and additional experiments are needed. However, for the device count rate, we can make definitive predictions based on the hotspot damping coefficient, ζ, of ref. [8] and our present work. For an NbN SNSPD fabricated on SiN, switching the substrate to GaN, should almost double β, which would allow us to reduce the device kinetic inductance by five times, while maintaining the same ζ. Thus a speedup of 5x without latching is expected. In addition, our work should help drive the choice of SNSPD material. For example, the acoustic properties of MoSi allow for a wider range of thermal boundary conductances than for NbN and WSi (see Supplementary Information). Thus MoSi may be the ideal material for examining the tradeoff between device speed and detection efficiency.

In summary, our method of extracting β is simple, the extracted values match those expected by acoustic modeling to a surprising degree, and our electro-thermal simulations give us a better understanding of the circumstances under which better accuracy can be expected. Simulations indicate that while our extraction of the β works extremely well, the extraction of the exponent n appears unreliable and may give a false impression of alternative physics if used without care. While similar measurements have been done before, the lack of comparison with theoretical expectation has hindered understanding and, in some cases, discrepancies with theory exist[21,23]. Additionally, work on SNSPDs and related detectors often use a linearized model of heat transfer which we have shown is not compatible with the data. Re-analysis of data in ref. [22] shows excellent agreement with our model (see Supplementary Fig. 4). Thus, we believe that superconducting nanowires such as those being created for high efficiency single-photon detection can be a convenient platform for probing nanoscale heat transfer phenomena, and that these investigations will yield improved detectors. Open questions about device performance trade-offs, such as that expected between detection efficiency and the maximum count rate as the TBC is varied, still need to be understood and experimentally realized to take full advantage of the promise of SNSPDs. Our work should help in answering those questions.

## Methods

Nanowire fabrication and measurement generally followed the approach set forth in our prior work[56,57], with some points of emphasis during the measurement. Fabrication of the devices measured for this work followed the same general methodology, but were fabricated by a variety of personnel over a period of approximately two years, with a variety of intentions for the fabricated devices. Few-nanometer-thick NbN films were reactively sputtered onto solvent cleaned and oxygen plasma-ashed substrates, after an in-situ plasma cleaning with Ar+. Electron beam lithography, using a 125 keV electron beam, and reactive ion etching in $CF_4$ was used to define NbN nanowires with widths and lengths ranging from 50 to 400 nm, and 3 to 100 μm respectively. Cryogenic measurements took place in one of two pulse-tube based closed cycle cryo-coolers with base temperatures ≲3 K. Care was taken to heat sink the chips to the cold-head, using GE-varnish under the chip, and at least ten aluminum wire bonds on the chip perimeter to improve the heat conductance to the cold head. Some of the chips were varnished on a grounded 'bed of nails' on a printed circuit board, while others were varnished directly to a gold plated OFHC copper mount. Apiezon-N grease was applied to any surfaces that were bolted together. Brass screws were used wherever possible to encourage firm contact between copper parts at low temperatures. The electrical measurement setup included an isolated voltage source in series with a bias resistor, which was attached to the DC port of a bias T, as depicted in the inset of Fig. 1a. The RF port was terminated with a 50 Ω termination and the DC&RF port was attached to the cryostat feedthrough attached to coax leading down to the measured device. The bias resistor was usually 10 kOhms, which was sufficiently small that the resulting load-line allowed us to resolve the hotspot plateau at low voltages for all devices.

### Reporting summary

Further information on research design is available in the Nature Research Reporting Summary linked to this article.

## Data availability

The $I_{hs}(T_b)$ data generated in this study are provided in the Supplementary Information. Additional data are available from the corresponding authors upon reasonable request.

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

## Acknowledgements

This work was supported by the DARPA Detect Project, grant no. DARPA ARO W911NF1620192. A.D. and J.A. were each supported by a NASA Space technology research fellowship, grant no's NNX14AL48H and NNX16AM54H respectively. D.Z. was supported by an A*STAR National Science Scholarship. The authors would like to thank Jim Daley and Mark Mondol who enabled our use of the Nanostructures Laboratory at MIT where the majority of the sample fabrication took place. We would also like to thank Donnie Keathley and Navid Abedzadeh for their help reviewing drafts of this manuscript, as well as all other members of the Quantum Nanostructures and Nanofabrication Group at MIT. Terry Orlando and Harvey Moseley provided invaluable feedback on this work.

## Author contributions

A.D. and K.K.B conceptualized these experiments. Fabrication of devices was done by A.D, M.O., M.C., R.B., J.T., Y.M., and I.F. Measurements were performed by A.D, D.Z., M.O., M.C., and Q.Z. Analysis and interpretation of the data was done by A.D, J.A, I.C., M.S., A.K., and K.K.B. Numerical simulations were performed by J.A with input from A.K. K.K.B. supported and supervised this work. A.D. prepared the manuscript with input from all co-authors.

## Competing interests

The authors declare no competing interests.
