## [Peer Review File · Nature Communications]

Self-Heating Hotspots in Superconducting Nanowires Cooled by Phonon Black-Body RadiationREVIEWER COMMENTS

Reviewer #1 (Remarks to the Author):

The manuscript in title of “Self-Heating Hotspots in Superconducting Nanowires Cooled by Phonon Black-Body Radiation” by A. Dane et al. reported a simple method to estimate the thermal boundary conductance (TBC) between the superconducting nanowire and the substrate by using the hotspot current (I_{hs}) of the nanowire. The authors calculated the effective phonon emissivity of NbN on various substrates by two different phonon mismatch model and the numerical results are in a good consistency with the experimental data.

However, most important frame works were done or mentioned by early reports, including the general thermal equation (Eq. 1), the approximate thermal equation (Eq. 2), and the phonon mismatch models for calculation. The value of the exponent n in equations had been reported to be strongly dependent on the materials of the superconducting nanowire and the substrate. The author focused on the NbN superconducting nanowire on various substrates. Though most of samples follows the model with exponent n of 4, so called “phonon black body radiation”, the results of a few of samples still deviate their model. Furthermore, as described in the manuscript, if the exponent n was treated as a fitting parameter, some of fitted values deviate from 4 significantly. These results implicate that the “phonon black body radiation” which has an exponent n of 4 might be suitable only for some cases with NbN superconducting nanowire on specific substrates and might not be able to apply for other systems.

By considering the requirement of high quality, I don't recommend to publish this manuscript on the Nature Communications. However, the result is good for other journals with some modifications.

Reviewer #2 (Remarks to the Author):

The manuscript by Andrew Dane et al reports on thermal boundary conditions that could affect performance of superconducting devices. This is a topic that has not yet been addressed in depth and the analysis presented is interesting because this works opens a route towards experimental determination of key parameters governing the thermal contact.

The analysis is thorough and interesting and compares a sufficiently wide range of substrate materials to support the claim of determining the thermal boundary conditions.

My main concern with the current manuscript is the use of a 3D (bulk) model for the phonon transport while the authors clearly indicate that this seems unphysical given a typical thermal phonon wavelength compared to the thickness of the detector. Is the good correspondence between data and model a coincidence? The interpretation of the data is mostly done on the basis of this, possibly wrong, microscopic model. This relates to the section on page 8 starting with 'All of our current data ... '

I have several additional comments that I hope the authors can address

A. The abstract may be misleading. The authors first write that thermal boundary conditions influence performance parameters of SSPDs. This statement I would agree with. They then

write that is important for all performance parameters and then weaken the claim in the introduction with 'is thought to affect quantum efficiency'

B. On page 1 'Energy deposited into the SNSPD is eventually released into the substrate in the form of phonons.' Suggests different processes occurring at different timescales. The quantum efficiency seems to be determined by processes at very short timescales, i.e. before phonon transport to the substrate.

C. Page 3. 'Skopkol' should be Skocpol

D. page 9. The use of R^2 as a measure for goodness of fit is uncommon and will be close to one even for a less than perfect fit because it is a squared quantity. This makes it difficult to interpret for non-experts. Is this an adjusted R^2 value? I would recommend using a different metric to quantify goodness of fit.

Reviewer #3 (Remarks to the Author):

The manuscript entitled "Self-Heating Hotspots in Superconducting Nanowires Cooled by Phonon Black-Body Radiation" discusses the in-depth analysis for the dynamics of hot-spot in the NbN superconducting nanowire single-photon detector. The authors considered that the thermal boundary conductance between the NbN nanowire and the substrate is the main component governing the heat dissipation from the hot spot in the operation of NbN SNSPD. Especially they argued that their thermal model gives high accuracy in the explanation of the experimental results.

In the manuscript, the authors report the results of the self-heating hot spot current measurements from 17 NbN nanowire samples with 6 different substrate materials, and the results are analyzed with the appropriate one-dimensional model and fittings to quantify the thermal boundary conductance (TBC).

The recent progress in the quantum computing fields requires the high performance of the single-photon detector measuring photons with quantum information. So the reviewer believes that the quantitative analysis on the NbN SNSPD is especially important nowadays.

The good thing about this manuscript is its in-depth analysis of the TBC. There might be no doubt in their report from the reviewer's point of view. However, there are some shortcomings, which hinder the broader impact on society as listed below.

= Authors use the one-dimensional model for the analysis. Then there should be clearly some limitations of their model assuming the fabricated NbN SCSPD as a one-dimensional wire because the real samples have definitely a finite width and thickness. So can authors determine what is the physical dimension that their model is valid?

= From the obtained results, is it possible to suggest the design rule to enhance the performance of NbN SCSPD such as the faster response or the increase of the measurement rate handling the high-speed detection for the successive single-photon inputs?

= Authors mentioned that their model is based on the time-dependent simulation. But the physical time scale doesn't seem to be given, and the scaled time-related parameters are discussed in Figure 3. Can the real-time scale be suggested on the basis of the model? And can the way of reducing the hot-spot removal time also be suggested? Or how does the engineered TBC affect the best performance of the state-of-the-art NbN SCSPD? The basic concern of the series of questions is related to the broader and practical impact of this work.

= The interface thermal boundary conductance is the main topic of study in this work. But in the final paragraph, the advantage of phononic crystal is argued, which does not seem to be relevant to this work. Also, the reason for discussing the aluminum having poor acoustic matching with the substrate is also not clear.

= Is the effect of etching damage in the NbN nanowire during the fabrication of the NbN SNSPD considered? What is the effective width of the superconducting region in the NbN nanowire, which justifies the one-dimensional model? Are the thermal properties at the boundary of etched NbN the same as the center of the NbN strip?

In conclusion, the study on the TBC using the NbN SCSPD is very impressive. However, the suggestion for the better performance of the NbN SCSPD with the new design rule is not provided, even though an accurate analysis on the TBC in NbN SCSPD is proposed.

Below is a point by point response to the items brought up during the review of our manuscript. Our responses are given in red. Again, thank you for your time, and the referees time in this matter.

Thank you,

Andrew

Reviewer #1 (Remarks to the Author):

The manuscript in title of “Self-Heating Hotspots in Superconducting Nanowires Cooled by Phonon Black-Body Radiation” by A. Dane et al. reported a simple method to estimate the thermal boundary conductance (TBC) between the superconducting nanowire and the substrate by using the hotspot current (I_{hs}) of the nanowire. The authors calculated the effective phonon emissivity of NbN on various substrates by two different phonon mismatch model and the numerical results are in a good consistency with the experimental data.

However, most important frame works were done or mentioned by early reports, including the general thermal equation (Eq. 1), the approximate thermal equation (Eq. 2), and the phonon mismatch models for calculation.

It's true that the general framework and equations were already shown to be useful for different systems. We did our best to acknowledge and cite those former works and would be happy to include any that we missed. However, the frameworks that we used have not yet been systematically applied to SNSPDs. SNSPDs are typically much thinner, narrower, and made from highly disordered materials and it was not clear to us at the beginning of this work how well these frameworks would apply. Most importantly, SNSPDs have become one of the primary devices for quantum optics experiments and low power optical communications. Knowledge produced using the devices themselves, not one-off experimental devices which might or might not be representative of SNSPDs, is invaluable for understanding and furthering device performance. Additionally, prior work that has been done in this direction appears to contain errors and erroneous conclusions regarding the thermal boundary conductance. This is due in large part to the fact that no paper that measured SNSPD-like devices, and very few relevant papers in general, compare measured TBC to expectations from theory. In short, the SNSPD literature is lacking a published article which correctly captures and explains this aspect of the devices.

For instance, Herr & Kadin 1996 (Johnson, M. W., Herr, A. M., & Kadin, A. M. (1996). Bolometric and nonbolometric infrared photoresponses in ultrathin superconducting NbN films.

Journal of Applied Physics, 79(9), 7069–7074. <https://doi.org/10.1063/1.361426>) used microwires to do a similar measurement and fitting to what we have here, except they included the erroneous linearization from Tinkham’s seminal paper. As a result, the value of the thermal boundary conductance they report is almost four times higher than what we get when we do a re-analysis in line with our paper (which is included in the supplementary information). Unfortunately, reference Herr et al is often cited. In at least one case the erroneous thermal boundary conductance of Herr seemed to be used as the starting point for one of two fitting parameters in a similar study on superconducting microwires. Despite an improved analysis in Stockhausen et al 2012 [Stockhausen, A., Ilin, K., Siegel, M., Södervall, U., Jedrasik, P., Semenov, A., & Hübers, H. W. (2012). Adjustment of self-heating in long superconducting thin film NbN microbridges. *Superconductor Science and Technology*, 25(3). <https://doi.org/10.1088/0953-2048/25/3/035012>], they appear to erroneously confirm Herr’s mistake.

Related works that attempted similar measurements do not relate their measurements to expectations based on acoustic theory, which makes it difficult to assess their results. For instance, during our revisions, we found that at least two manuscripts, Stockhausen et al and Schmidt et al 2017 [Schmidt, E., Ilin, K., Siegel, M., & Preparation, A. S. (2017). AlN-Buffered Superconducting NbN Nanowire Single-Photon Detector on GaAs, 27(4), 10–14] which report thermal boundary conductances that appear too large to be explained by AMM or DMM theory. A paper that the field of SNSPDs can point to as a reference and tool to avoid this kind of situation is sorely needed. Our paper not only makes the relevant comparisons, but we have calculated a wide range of useful superconductor-substrate pairs and provided guidance in this regard.

In our case, comparison to the existing theory was essential, because we started this work thinking that we would need to alter the description of our devices to account for the few-nm thickness, an issue that is not encountered in devices made from thicker materials, or which is completely ignored. Much to our surprise we did not need to do that in order to get excellent agreement between our data and predictions by acoustic mismatch.

Revisions made to our manuscript in response to the above point:

- Added statements to help differentiate the current work from what has been done in the past.
- Added statements to emphasis the erroneous nature of the Herr et al paper and the follow up work by Stockhausen et al.
- Added statements that point out that references typically do not compare with expectations from theory.
- Added statements to point out references which imply a larger than possible thermal boundary conductance.

The value of the exponent n in equations had been reported to be strongly dependent on the materials of the superconducting nanowire and the substrate. The author focused on the NbN superconducting nanowire on various substrates. Though most of samples follows the model with exponent n of 4, so called “phonon black body radiation”, the results of a few of samples still deviate their model. Furthermore, as described in the manuscript, if the exponent n was treated as a fitting parameter, some of fitted values deviate from 4 significantly. These results implicate that the “phonon black body radiation” which has an exponent n of 4 might be suitable only for some cases with NbN superconducting nanowire on specific substrates and might not be able to apply for other systems.

This is a good point, and one which we will incorporate into the current manuscript better. We still believe that calling this ‘phonon black body’ radiation will be the best for helping to resolve the issues within the field, but we fully agree that it does not apply to all cases. Because the seminal works in the field adopted a linearized description of the thermal properties of the boundary, that description has propagated and given some researchers the false impression of exclusively diffusive heat transfer. From our perspective, the first assumption should be this phonon black body description, and the exceptions should be in cases where the substrate material pushes towards the diffusive limit. In follow up work we will more fully examine the case of NbN on PET plastic, which creates both a large interfacial thermal resistance, and has a limited thermal conductance in its bulk.

Revisions made to our manuscript in response to the above point:

-We have attempted to further explain the nature of n in the exponent, and we have tried to emphasize that our method may do a poor job of extracting n , based on our simulation work where we fixed the relevant n values, only to have the fitting return an n value that was not programmed in.

-We have added references that discuss the changing nature of n depending on the phonon dimensionality.

By considering the requirement of high quality, I don’t recommend to publish this manuscript on the Nature Communications. However, the result is good for other journals with some modifications.

These comments have been very helpful, and addressing them has make our manuscript much better. Of course, we respectfully disagree with the assessment that our paper should not be published in Nature Communications. On the contrary, we think that researchers in the field of SNSPDs will welcome this publication. Thank you!

Reviewer #2 (Remarks to the Author):

The manuscript by Andrew Dane et al reports on thermal boundary conditions that could affect performance of superconducting devices. This is a topic that has not yet been addressed in depth and the analysis presented is interesting because this work opens a route towards experimental determination of key parameters governing the thermal contact.

The analysis is thorough and interesting and compares a sufficiently wide range of substrate materials to support the claim of determining the thermal boundary conditions.

My main concern with the current manuscript is the use of a 3D (bulk) model for the phonon transport while the authors clearly indicate that this seems unphysical given a typical thermal phonon wavelength compared to the thickness of the detector. Is the good correspondence between data and model a coincidence? The interpretation of the data is mostly done on the basis of this, possibly wrong, microscopic model. This relates to the section on page 8 starting with 'All of our current data ...'

When we began this work, we were looking for a more unusual explanation of the system at hand, expecting that the phonon system of such a thin film would not be described by the 3D model that we ultimately used. However, the agreement between this model and the data, over a wide variety of substrate types, gives us confidence that this model captures the essential aspects of the devices. However, we realize that our model simplifies a very complex situation. This is good for engineers who want to know how to make faster devices. But, for any given substrate/film combination, we may need to know almost atomic level details of the interface to correctly model it, which we believe is well outside of the scope of this work. In addition, the correct accounting for the spectrum may require considering Raleigh-Lamb waves while simultaneously describing how this effects electron-phonon coupling. In the simplest case, modifying the density of states results in a slightly over heated phonon temperature, relative to what it would be with the standard 3D density of states. The higher temperature means shorter phonon wavelengths, potentially making the 3D picture once again valid. This overheating might be too small for us to 'see' with our method, but the error that it would create in β appears to push the distribution of Fig. 2c towards being more centered on one. This seems too fortuitous, and in a paper that relies heavily on simplifying assumptions, it may be better to leave this detail to future work.

Revisions made to our manuscript in response to the above point:

-We have made reference to models that can account for size effects in phonon systems. While the simplest model would actually help our argument, realistic models would require much more detailed work, simulations and calculations which we believe are outside of the scope of the present work.

I have several additional comments that I hope the authors can address

A. The abstract may be misleading. The authors first write that thermal boundary conditions influence performance parameters of SSPDs. This statement I would agree with. They then write that is important for all performance parameters and then weaken the claim in the introduction with 'is thought to affect quantum efficiency'

Sorry about this. We believe that the thermal boundary conductance affects each of the parameters. Especially as it has become clear about how Fano fluctuations and energy loss during the down conversion cascade suggests that phonon escape and thermal boundary conditions play an important role in determining the quantum efficiency, the width of the PCR curve, and the intrinsic timing jitter.

Revisions made to our manuscript in response to the above point:

-We revised the language about what parameters are effected by TBC.

B. On page 1 'Energy deposited into the SNSPD is eventually released into the substrate in the form of phonons.' Suggests different processes occurring at different timescales. The quantum efficiency seems to be determined by processes at very short timescales, i.e. before phonon transport to the substrate.

Again here, Fano fluctuations are an important aspect to consider. Even at the earliest stages of the photo-detection process, energy from the photon may be split into energy in the electron subsystem and phonon subsystem. This energy can be traded back and forth. However, energy in the phonon sub-system has a chance to escape, reducing the energy that is available to disrupt superconductivity. This chance is determined by the thermal boundary conductance, but in this context would often be described as the 'phonon escape time'. Of course, you are right that there is another aspect of things. After the breakdown of superconductivity and the creation of a joule heated hotspot, it is vitally important that this hotspot cool down in such a way that the device does not 'latch' into a stable resistive state. This aspect of the thermal boundary conductance comes in at times much later than those relevant for fano fluctuations.

Revisions made to our manuscript in response to the above point:

-We have added explicit numbers related to the time processes involved during the detection process, however we should emphasize, the measurements in this paper are DC (time independent) measurements. Never-the-less, the parameters estiated in our work can be used to model the wire behavior relevant for detection.

-In parts of the manuscript we've tried to re-emphasize the connection with our work and the detector processes

C. Page 3. 'Skopkol' should be Skocpol

Corrected, thank you!

D. page 9. The use of R^2 as a measure for goodness of fit is uncommon and will be close to one even for a less than perfect fit because it is a squared quantity. This makes it difficult to interpret

for non-experts. Is this an adjusted R^2 value? I would recommend using a different metric to quantify goodness of fit.

Sorry about this. The quoted values are adjusted R squared values, generated automatically by MATLAB while using its curve fit toolbox. We hope this is enough to satisfy this point, and if not we would be happy to calculate an alternative. In addition, in the supplementary materials we provide plots of data and fits to allow readers to judge the goodness of fit ‘by eye’.

Revisions made to our manuscript in response to the above point:

- We changed the language to reflect adjusted R-square.

Thanks to Reviewer #2 for honest and helpful remarks!

Reviewer #3 (Remarks to the Author):

The manuscript entitled “Self-Heating Hotspots in Superconducting Nanowires Cooled by Phonon Black-Body Radiation“ discusses the in-depth analysis for the dynamics of hot-spot in the NbN superconducting nanowire single-photon detector. The authors considered that the thermal boundary conductance between the NbN nanowire and the substrate is the main component governing the heat dissipation from the hot spot in the operation of NbN SNSPD. Especially they argued that their thermal model gives high accuracy in the explanation of the experimental results.

In the manuscript, the authors report the results of the self-heating hot spot current measurements from 17 NbN nanowire samples with 6 different substrate materials, and the results are analyzed with the appropriate one-dimensional model and fittings to quantify the thermal boundary conductance (TBC).

The recent progress in the quantum computing fields requires the high performance of the single-photon detector measuring photons with quantum information. So the reviewer believes that the quantitative analysis on the NbN SNSPD is especially important nowadays.

The good thing about this manuscript is its in-depth analysis of the TBC. There might be no doubt in their report from the reviewer’s point of view. However, there are some shortcomings, which hinder the broader impact on society as listed below.

= Authors use the one-dimensional model for the analysis. Then there should be clearly some limitations of their model assuming the fabricated NbN SCSPD as a one-dimensional wire because the real samples have definitely a finite width and thickness. So can authors determine what is the physical dimension that their model is valid?

There are two lengths scales to consider, both of which enable a 1-D model for our nanowires. Firstly, the thickness and width of our nanowires is much less than the effective penetration depth, or pearl length, which ensures that currents flowing in the nanowire are almost completely uniform (the current density is constant across the wire cross section). The pearl length is the penetration depth squared divided by the thickness. For NbN, this is roughly 8 μm which is much larger than the few nm thickness or ~ 100 nm width of the wires. This means we don't have to consider what happens when the current density is higher towards the edges of the film than it is in the middle, as would likely happen if the wire width began approaching the pearl length.

Perhaps more pertinent here is the thermal healing length, which is the length scale that governs the variation in temperature inside of the wire [Skocpol, W. J., Beasley, M. R., & Tinkham, M. (1974). Self-heating hotspots in superconducting thin-film microbridges*. *Journal of Applied Physics*, 45(9), 4054–4066. <https://doi.org/10.1063/1.1663912>]. We've estimated the thermal healing lengths for the wires we have measured ranges from ~ 100 nm to 850 nm, with a median value of 136 nm. Therefore, our wires are in the limit where the thickness is much less than the thermal healing length, but the width is similar. In a handful of cases, the wire width is less than the estimated thermal healing length, but of the same order of magnitude.

Revisions made to our manuscript in response to the above point:

-We've added language that addresses whether our nanowires are really 1D from a thermal perspective.

-We've added language that mentions the pearl length and whether our device is 1D from an electrical perspective.

-We've added our estimate of the thermal healing length for each sample, to the supplementary information.

= From the obtained results, is it possible to suggest the design rule to enhance the performance of NbN SCSPD such as the faster response or the increase of the measurement rate handling the high-speed detection for the successive single-photon inputs?

One thing that we're trying to point out in our paper is that this aspect of the devices is understudied relative to its importance. So, for many of the device parameters, even though we know they will be affected by the magnitude of the TBC, we don't necessarily know to what extent. In addition, at the boundaries of device performance, there should be trade-offs that are not yet elucidated. For instance, if the TBC could be arbitrarily varied, we expect that increasing it would allow use to count photons more quickly, because the device could cool and reset faster. However, if the TBC were high enough, energy lost to the substrate during the initial photo-

detection could lead to reduced quantum efficiency. So, a trade-off exists, but when it becomes relevant, and what the TBC would need to be is still far from known.

However, we can still make predictions about the speed of the device based on the work of Kerman et al [Kerman, A. J., Yang, J. K. W., Molnar, R. J., Dauler, E. A., & Berggren, K. K. (2009). Electrothermal feedback in superconducting nanowire single-photon detectors. *Physical Review B*, 79(10), 100509. <https://doi.org/10.1103/PhysRevB.79.100509>] combined with our measurements. We have included this and given some guidance. For instance, by making the switch from a substrate that has low TBC with NbN, like SiN, to a substrate with higher TBC with NbN, we would expect a speed up of 5x. Additionally, some SNSPD wire metals have advantages over others. MoSi for instance can achieve much higher TBC with the basket of common substrates that we examined.

Revisions made to our manuscript in response to the above point:

-We've added language to show how, in the Kerman framework, changing the thermal boundary conductance by changing the substrate can lead to substantial increases in speed, by allowing one to reduce the overall device kinetic inductance. In general, we indicate how the 'hotspot damping coefficient' can be used in this regard.

-We now mention that some materials choices provide greater opportunities for thermal engineering of these devices. MoSi in particular stands out as a well-demonstrated SNSPD material that allows for a much greater range of TBCs than competing NbN and WSi.

-We've added language to the effect that there are trade-offs that still need to be elucidated.

= Authors mentioned that their model is based on the time-dependent simulation. But the physical time scale doesn't seem to be given, and the scaled time-related parameters are discussed in Figure 3. Can the real-time scale be suggested on the basis of the model? And can the way of reducing the hot-spot removal time also be suggested? Or how does the engineered TBC affect the best performance of the state-of-the-art NbN SCSPD? The basic concern of the series of questions is related to the broader and practical impact of this work.

I think it's understood, but just to be clear, we used a model that we have employed previously, where the time dependence was a necessity, but here, the experiments are done at DC, and all time dependence in the model is removed (all d/dt terms equal to zero). The time scales used in the axis are consistent with a large swath of simulations and prior work, however we can try to make it easier to interpret by giving some physical times.

We expect that choosing substrates to maximize the thermal boundary conductance, based on our present work, will in general reduce the time it takes for a hotspot to cooldown after photo-detection, however when we start to ask about the time dynamics of the hotspot during photodetection, things quickly become complex to the level that they require much more consideration and modelling than would be appropriate for this work (see for example Vodolazov 2017 referenced in our manuscript). At later times, the electrothermal feedback between the growing hotspot and the readout circuit become the dominant consideration. Again,

the thermal boundary conductance has a role to play, but the conditions of the measurement are important, making it difficult to provide a simple picture without extensive explanation.

The statement we might be able to make without too many caveats has to do with the device speed. SiN is a substrate material that we commonly use, and it is on the lower end of TBCs compared to substrates that we examined. If we used GaN instead, the TBC would almost double. In that case, we could expect to be able to reduce the kinetic inductance by more than 5 times, while still maintaining the same ‘hotspot damping coefficient’. All else being equal this could translate into devices that are more than 5 times faster while using standard 50 Ohm readout circuitry, potentially pushing us into GHz counting territory.

Revisions made to our manuscript in response to the above point:

-We’ve added physical times to manuscript to make it easier to think about different regimes of the photodetection process. We’ve tried to reiterate that the phonon escape time is another way to describe the thermal boundary conductance, but from a different perspective.

-We’ve tried to make it more clear to device designers what to do to make faster devices.

= The interface thermal boundary conductance is the main topic of study in this work. But in the final paragraph, the advantage of phononic crystal is argued, which does not seem to be relevant to this work. Also, the reason for discussing the aluminum having poor acoustic matching with the substrate is also not clear.

We were trying to tie our work into some other relevant subjects, but clearly these topics were a little far afield. We’ve removed them to make room for discussion of some of the foregoing points.

Revisions made to our manuscript in response to the above point:

-We removed references to Al and phononic structures.

= Is the effect of etching damage in the NbN nanowire during the fabrication of the NbN SNSPD considered? What is the effective width of the superconducting region in the NbN nanowire, which justifies the one-dimensional model? Are the thermal properties at the boundary of etched NbN the same as the center of the NbN strip?

We do not try explicitly to incorporate damage to the edges in our model, except that we use the experimentally measured resistance in the normal state when we use formula #2 in the main text when extracting beta from the fit. We sometimes see that the total resistance of a device is larger than the expected resistance based on the sheet resistance measured on the unpatterned film, times the # of squares from the device design. This increase in resistance has a few potential sources: (1) over/under exposure of the resist can lead to a device that is slightly wider/narrower than expected, which changes the # of squares. (2) There may be variations in the material resistivity on the scale of the nanowire width that become pronounced when the device is patterned, but are averaged out when measuring the film. (3) Chemical or physical damage to the edges of the NbN due to fabrication. There is some reason to believe that, depending on the fabrication process, edge damage may occur, and we cannot strictly rule it out.

However, one of the strengths of the MIT group in particular has been the fabrication of high efficiency devices. Internally, we sometimes attribute the good performance of our devices to careful and steadily improving fabrication processes over time. For instance, we have known for some time that resist developers which contain TMAH (which is extremely common) are able to attack Nb containing compounds, so we have sought to reduce exposure to this chemical or eliminate its use altogether. Some resists appear to allow us to maximize the apparent critical current density, which is one reason we use ZEP: it has a higher etch resistance in reactive ion etching, and it appears it may limit edge damage for that reason.

In the simplest case, edge damage could render the NbN non-conducting, effectively making the metal wire narrower while replacing it with some dielectric. This would have to occur in such a way that it is not revealed under SEM imaging, otherwise we would notice it during routine examination of the devices during fabrication. If this were the case, then it means in our fits to determine Beta, we would be using a width that was too large, and that we would be underestimating beta. If this was true among all of our devices, it could be a reason why our beta_fit is systematically lower than calculation. If all of our widths were uniformly 16% lower than what we expect, the average value of Bfit/BAMM in figure 2.c would be equal to one. That is, a reduced width improves our model. However, it is not clear that such a percentage reduction in width should occur. Perhaps if the width were effectively reduced by a diffusion process coupled with a variable temperature due to heating during etching. Otherwise, it seems likely that a constant offset would occur, say due to ion bombardment that can reach the film by penetrating the bottom corner of the resist at an angle.

Revisions made to our manuscript in response to the above point:

- We have added language, to address whether the wire is one dimensional.
- We've added a reference about edge effects in NbN wires, however we believe a combination of: (1) existing good fabrication processes that avoid edge damage, (2) using measured R and width values helps us avoid major issues related to this point.

In conclusion, the study on the TBC using the NbN SCSPD is very impressive. However, the suggestion for the better performance of the NbN SCSPD with the new design rule is not provided, even though an accurate analysis on the TBC in NbN SCSPD is proposed.

Thank you for your kind words and useful critiques. We've tried to improve the manuscript based on your suggestions. Thank you!

REVIEWER COMMENTS

Reviewer #2 (Remarks to the Author):

I have read the response of the authors to all referee comments and I believe that the response to all points raised is adequate. I confirm that the changes made to the manuscript are appropriate and address all comments. My opinion is that the manuscript should be published in its current form.

The current work is a significant step in the investigation of thermal boundary conditions. I believe that a deeper understanding of the thermal boundary conditions is perhaps the most important issue to create a faster and better detector technology. Besides the technological importance the deeper understanding of these processes are of fundamental importance. The field is strongly driven by theory and this paper provides important experimental data.

Reviewer #3 (Remarks to the Author):

Many answers to my former questions are satisfactory, but still, there is a concern as described below :

The role of TBC especially for the SNSPD is important not only for the breaking of Cooper pairs by the absorption of a single photon, but also the fast removal of heat generated by quenching. But the experiment is performed in the DC, not the pulse scheme, and such a DC scheme seems to be considered again in the simulation results, which reduces the impact of this work even though the work itself is self-consistent.

For all other concerns that I raised, the detailed explanations were provided in the author's responses.

Dear Editor and Referees:

We would once again like to thank you for your time and input. Below we attempt to respond to the additional points brought up by the referees after we submitted our revised manuscript.

Thank you,

Andrew Dane and coauthors.

Responses:

Reviewer #1:

No response given.

Reviewer #2:

No points requiring response.

Reviewer #3:

“The role of TBC especially for the SNSPD is important not only for the breaking of Cooper pairs by the absorption of a single photon, but also the fast removal of heat generated by quenching. But the experiment is performed in the DC, not the pulse scheme, and such a DC scheme seems to be considered again in the simulation results, which reduces the impact of this work even though the work itself is self-consistent.”

Our response:

While we agree that additional measurements and characterizations are useful and interesting, we believe our work will be broadly applicable for those researching SNSPDs for a number of reasons which we summarize here and explain further below:

1. The DC measurement scheme that we propose is one that could easily be implemented by any researcher studying SNSPDs, because it doesn't require additional equipment beyond what is typically used to measure SNSPDs, and it is agnostic to the design of the device.
2. We would argue that low frequency pulsed electrical measurements are not significantly different from the ones that we made, and that higher frequency measurements that are qualitatively different, would require additional equipment, experimental design, and likely devices that are purpose built for this type of measurement. These things therefore could make pulsed electrical measurements less accessible and potentially less impactful.

3. There are some pulsed optical measurements that could be useful, however as with the first two points, these types of measurements would require purpose built experiments and/or devices, rather than being useful for any and all SNSPD experiments.

Because SNSPD detection efficiency is a function of the DC bias current, almost all SNSPD measurement setups are equipped to vary the DC bias current through a nanowire detector and measure the DC voltage across the nanowire. Measuring the IV curve of the device at the base temperature of the cooling system is a standard measurement that we take on virtually every device that we measure. What we are suggesting in our paper, is that by taking these measurements at a few temperatures, and by correctly modelling the system (which hasn't been done for SNSPDs in the past), that SNSPD researchers can get a measure of the thermal boundary conductance for their specific nanowire and substrate, without doing anything that they aren't already setup to do. We believe that this makes our approach broadly applicable.

We have sometimes taken our IV curves in a low frequency AC fashion (frequencies < 1 kHz), typically so that we could measure switching current distributions and switching rates. While we did not study that here, and it requires a different setup than what we typically used to measure SNSPDs, anecdotally we never saw anything particularly different in the resulting IV curves. One possible difference is that in a low frequency pulsed measurement scheme, we can attempt to reduce the duty cycle and minimize the average power dissipated on the sample. However, we have not seen evidence that this is necessary for our devices. Further, our standard mounting procedure is intended to provide good thermalization. This is documented in our methods, but mounting the sample to gold coated OFHC copper, using thermal grease and numerous wire bonds (for good electrical and thermal contact).

In order to make pulsed electrical measurements that are qualitatively different from what we describe above, we would need to provide pulses and monitor voltages at frequencies that approach the frequencies of the processes that we are interested in probing. For instance, the phonon escape times may be on the order of 100 ps, which corresponds to a frequency of 10 GHz. SNSPDs usually have significant kinetic inductance, and depending on the material and device geometry, the total inductance of device could easily be between 10 and 1000 nanoHenries. At 10 GHz, SNSPDs are significantly mismatched from the standard 50 Ohm impedance of RF electronics, so efficient coupling of an RF pulse to the device would require impedance matching to the particular device. After switching, the voltage response of the device would be dominated by the large inductance and any impedance matching, not unlike how standard SNSPD pulses are dominated by the inductive time constants that govern the rise and fall time of the pulse. Any effects from the thermal boundary conductance (which could contribute to the size of the hotspot resistance) would be subtle and would need to be extracted with care. In short, this type of measurement requires the RF infrastructure, careful device design, and data extraction to make it work.

Alternatively, you might ask about pulsed optical measurements. Time domain thermo-reflectance would be an interesting additional measurement to make, though in our preliminary work the need for additional metal layers to transduce the signal could interfere with the dimensionality considerations and superconducting properties of the NbN. Alternatively, as we alluded to in the

previous paragraph, when the device fires from a single photon event, the output pulse may contain some clues about the TBC, however they would be subtle and would require large bandwidths to resolve, and it is not clear what accuracy the extracted information would have. Relatedly, if we wanted to understand the smallest electrical time constant at which latching is avoided, and thus understand something about the TBC from the perspective of Kerman et al , we would need to create and measure a set of devices with a range of kinetic inductances, or create a purpose built devices with a variable resistance.

In short, we believe our scheme is broadly applicable because it doesn't require additional infrastructure or purpose built devices. Instead, anyone measuring an SNSPD can follow our work and understand something about the thermal boundary conductance of their device/substrate.

Actions taken:

We have added some additional language to the manuscript emphasize that our method does not require additional infrastructure or custom designs to work.

REVIEWERS' COMMENTS

Reviewer #3 (Remarks to the Author):

The authors properly answered what I was concerned about. I agree with the publication of this manuscript in its current form.